# Psychosis in Women: Time for Personalized Treatment

**DOI:** 10.3390/jpm11121279

**Published:** 2021-12-02

**Authors:** Marianna Mazza, Emanuele Caroppo, Domenico De Berardis, Giuseppe Marano, Carla Avallone, Georgios D. Kotzalidis, Delfina Janiri, Lorenzo Moccia, Alessio Simonetti, Eliana Conte, Giovanni Martinotti, Luigi Janiri, Gabriele Sani

**Affiliations:** 1Department of Geriatrics, Neuroscience and Orthopedics, Fondazione Policlinico Universitario A. Gemelli IRCCS, Università Cattolica del Sacro Cuore, 00168 Rome, Italy; giuseppemaranogm@gmail.com (G.M.); avallonecarla@yahoo.it (C.A.); giorgio.kotzalidis@gmail.com (G.D.K.); delfina.janiri@gmail.com (D.J.); lorenzomoccia27@gmail.com (L.M.); alessio.simo@gmail.com (A.S.); emanuele.caroppo@aslroma2.it (E.C.); luigi.janiri@unicatt.it (L.J.); gabriele.sani@unicatt.it (G.S.); 2Institute of Psychiatry and Psychology, Department of Geriatrics, Neuroscience and Orthopedics, Fondazione Policlinico Universitario A. Gemelli IRCCS, Università Cattolica del Sacro Cuore, 00168 Rome, Italy; 3Department of Mental Health, Local Health Unit Roma 2, 00182 Rome, Italy; emanuelecaroppo@gmail.com; 4National Health Service, Department of Mental Health, Psychiatric Service for Diagnosis and Treatment, Hospital G. Mazzini, Azienda Sanitaria Locale 4, 64100 Teramo, Italy; domenico.deberardis@aslteramo.it; 5Department of Neurosciences, Imaging, and Clinical Sciences, University G. D’Annunzio, 66100 Chieti-Pescara, Italy; giovanni.martinotti@gmail.com; 6Department of Neurosciences, Mental Health, and Sensory Organs (NESMOS), Sapienza University of Rome, 00189 Rome, Italy

**Keywords:** women, psychosis, schizophrenia, gender differences, personalized treatment

## Abstract

Early detection and prompt treatment of psychosis is of the utmost importance. The great variability in clinical onset, illness course, and response to pharmacological and psychosocial treatment is in great part gender-related. Our aim has been to review narratively the literature focusing on gender related differences in the psychoses, i.e., schizophrenia spectrum disorders. We searched the PubMed/Medline, Scopus, Embase, and ScienceDirect databases on 31 July 2021, focusing on recent research regarding sex differences in early psychosis. Although women, compared to men, tend to have better overall functioning at psychotic symptom onset, they often present with more mood symptoms, may undergo misdiagnosis and delay in treatment and are at a higher risk for antipsychotic drug-induced metabolic and endocrine-induced side effects. Furthermore, women with schizophrenia spectrum disorders have more than double the odds of having physical comorbidities than men. Tailored treatment plans delivered by healthcare services should consider gender differences in patients with a diagnosis of psychosis, with a particular attention to early phases of disease in the context of the staging model of psychosis onset.

## 1. Introduction

It is widely recognized that prolonged duration of untreated psychosis predicts worse symptoms and poorer social functioning and quality of life. For this reason, early detection and prompt treatment of psychosis is of the utmost importance [1]. It is widely recognized that schizophrenia and first-episode psychosis may show great variability in clinical onset, illness course, and response to pharmacological and psychosocial treatment [2]. Some aspects of this observed heterogeneity may be in great part gender-related.

In fact, there is sexual dimorphism in the development of brain areas [3,4] and this results in differences in the final volumes of gray and white matter [5,6,7]. This in turn results in the development of differential emotional skills between the two genders [8]. Sexual dimorphism affects the function of the dopaminergic system [9], which is known to be one of the central physio-pathogenetic mechanisms of schizophrenia and the psychoses in general [10,11,12,13]. Taken together, these studies point to differential characteristics of male and female brains right from the beginning of psychosis [14], thus to differential response to pharmacotherapy [15]. Therefore, reconsidering the evidence, the need for personalized, gender-related care emerges; this is stressed by recent studies [16,17,18].

There is a growing interest in current research focusing on several characteristics of psychotic illness separately analyzed in male and female populations. Subtle differences have been reported in many areas, with considerable implications for treatment [18]. Women may be disadvantaged as concerns access and quality of care for psychosis due to several factors, among which the fact that the emergence of psychosis is brought to clinical attention earlier for men than for women. Besides this, women have greater metabolic and endocrine-induced antipsychotic side effect risks. According to the estrogen hypothesis, estrogens raise the vulnerability threshold for the outbreak of the illness, so women would be protected against schizophrenia between puberty and menopause to some extent by their relatively high gonadal estrogen production during this time. This could explain the fact that women often have a later age of onset and could justify the belief that some therapeutic treatments associated with estrogens could be useful for improving symptoms and cognition, especially in women. According to the neurodevelopment hypothesis, it seems that women need more risk factors (such as family risk and life events) in order to develop schizophrenia than men, and this seems to be confirmed by the finding that usually men seem to present with a more deteriorated profile than women before the onset of the illness [2]. The existence of two distinct schizophrenias, masculine and feminine, has also been hypothesized. They would correspond to different forms of expression of specific psychotic symptoms and a different expression of psychotic illness due to the differential hemispheric organization, more symmetrically organized and less lateralized in the female compared to the male brain [19]. Some researchers have stressed the necessity to focus on the potential influence of the hypothalamic–pituitary–gonadal axis in the development of psychosis in women. In schizophrenia, an increased incidence has been shown in periods of low estradiol concentrations. Many women with schizophrenia, even in the untreated prodromal phase, experience estradiol deficiency and gonadal dysfunction, which might have put them at increased risk and might be due to stress-induced hyperprolactinemia [20]. It could be speculated that stress, which can induce hyperprolactinemia, has a stronger effect on women than on men in emerging psychosis [21,22]. Recently, central nervous system autoimmunity has been implicated in the etiology of psychosis, with specific gender differences [23]. It has been suggested that prolactin may underlie the excess of morbidity and early mortality in naïve patients with a first episode of psychosis through a specific pathway that intertwines inflammatory, immune and metabolic trajectories [24].

It can be argued that all possible models can contribute to explain gender differences in the predisposition to psychosis and in the expression and progression of illness. In people at high-risk of psychosis, differences between men and women in the expression of illness extend across a continuum, from the subclinical forms of illness to the debut of psychosis [14]. There are many special issues for women affected by psychosis that deserve particular attention, from sexuality (in terms of relationships, contraception, sexually transmitted diseases or sexual victimization) to the peripartum period (pregnancy, childbirth, postpartum period), from prescription of antipsychotic drugs to parenting (women with psychosis become parents much more frequently than men) and menopause [18].

To assess the need for personalized, gender-related interventions in the psychoses, especially schizophrenia, we conducted a literature search focusing on gender-related care for schizophrenia and related spectrum disorders. Given the importance of the presence of differential characteristics at the start of a clinical course [14], which points to the need to avoid delays in care, we focused on the first episode of psychosis (FEP).

## 2. Materials and Methods

We searched the PubMed/Medline, Scopus, Embase, ScienceDirect databases, since their inception, on 24 November 2021, using the terms (“schizophrenia”(ti) OR “early psychosis”(ti) OR “psychosis”(ti) OR “psychotic”(ti)) AND (“women”(ti) OR “female”(ti) OR “gender difference*”(ti) OR “sex difference*”(ti)), yielding 981 records. We also searched the ClinicalTrials.gov site for further articles. Inclusion criteria included original studies in peer reviewed journals focusing on sex differences in first-episode cohort treatment and outcome. Both longitudinal and cross-sectional studies were eligible and could be retrospective or prospective. There were no time or language limits in regard to the selection of appropriate studies. Reviews or meta-analyses focusing on early psychosis in female patients were not included, but their reference lists were consulted to identify further eligible studies. However, reviews were included if they had effectively summarized preceding evidence comprehensively. Studies were excluded if they did not focus on or were unrelated to the subject matter, if they were case reports or series, or opinions (e.g., letters to the editor or editorials) with no original data. All these were grouped under the heading Opinions, and reviews and metanalyses were grouped under the heading Reviews. Excluded were also animal studies, studies focusing on in vitro differences, those with inadequate study designs and those not reporting data, such as protocols and studies in the ClinicalTrials.gov database which were recruiting and had not concluded data gathering. A particular attention was paid to publications of the last five years (from 2017 to 2021) due to the generally increased quality of the studies published during this time period.

In our search, we adopted the Preferred Reporting Items for Systematic Reviews and Meta-Analyses (PRISMA) statement [19] to explore the possibility of a systematic review and meta-analysis. The latter two were not possible due to the methodological heterogeneity of eligible studies, hence we opted for a narrative review. The result of our search is shown in the signal flow diagram pictured in Figure 1.

## 3. Results

The above mentioned strategy yielded an output of 978 records. By applying inclusion/exclusion criteria, there were 17 articles left to comment on (Table 1). Figure 1 shows the selection process and the inclusion of eligible studies, along with the reasons for exclusion of the discarded articles. The first title appearing in the search was from April 1949, the last was published on 7 September 2021. The first included study was published online in September 2015, but published in a definitive form in December 2017 [25], and the most recent appeared online on 6 June 2021 [26]. The frequency of published studies on this subject has intensified across the years. Suffice it to look at Table 1; five of the 17 finally included articles were published in 2021, even before the year was over.

It is no surprise that the clinical presentation of psychosis is affected by gender, due to the different developmental trajectories between the two genders and their specific brain differentiation [20]. This differentiation is also at the basis of gender-specific presentations of other psychiatric disorders [21]. When dimorphism affects brain structure and function, it is also possible that the function of peripheral systems is also affected. One such system is the endocrine, and investigators have focused on the pituitary hormone prolactin, which is part of the stress circuitry. Results in the search of hyperprolactinemia as a marker of psychosis have been conflicting. The levels of serum prolactin were found to be elevated [22], not dissimilar [23], or even reduced in women compared to men [27] in the included studies; hence the need for greater personalization when taking care of patients with psychosis. Another such issue regards the immune system, which like the endocrine and the nervous systems regards cell–cell communication and uses similar mechanisms to carry out its functions. Neuroimmune differences affect response to treatment in the two principal human genders, but the data heretofore collected are far from being complete [24].

Gender generally had a considerable influence on the nature of the clinical presentation of psychosis. Male patients showed greater risk for substance use than female patients and, during first episode psychosis, men more frequently show greater negative and disorganization symptom burden, while women present with more affective and anxiety symptoms, and risk for parasuicidal behavior [28]. It has been shown that gender differences at presentation are independent of age and ethnicity. Furthermore, there is evidence that after adjusting for illicit substance use, negative symptoms still remain more prominent in men while manic and depressive symptoms became even more prominent in women, thus suggesting the idea that gender differences in the clinical presentation of psychosis and in symptom domains may reflect specific and identifiable sex differences in pathophysiology [29].

It has been largely demonstrated that delay in receiving effective treatment for psychosis adversely impacts outcomes. Gender stereotypes may negatively influence the first service contact and specialized care for psychosis both for women and men, although in different ways. Women may be considered overdramatic by old-fashioned, poorly trained male assessors and their symptoms may be often underestimated and perceived as an attempt to gain attention, while men may have difficulties in disclosing psychotic symptoms or asking for help and may be perceived as more dangerous due to their physicality [30]. Ferrara and Srihari [17] recently separately described factors affecting access to care (age at onset, clinical presentation, pathways to care, and gender roles) and factors affecting quality of specialized care for women at first-episode psychosis (sexual and reproductive health, preventive medicine, and trauma). Longer delay in receiving care for female patients may be determined by the onset of psychosis later in life, and by the fact that women tend to present with more mood symptoms compared to men, and may receive a misdiagnosis of a primary affective disorder instead of psychosis. Moreover, women with psychosis may need more support and education in order to prevent sexually transmitted diseases and unintended pregnancies and should be carefully directed to available preventive screening (Pap test, mammography etc.). Women deserve a particularly attentive support to evaluate the impact of prescribed antipsychotic medication, both in regard of possible side effects (weight gain, hair loss), and for potential teratogenic risks and results on reproductive health (e.g., secondary amenorrhea due to hyperprolactinemia) [17].

Some authors have claimed that men with a longstanding psychotic disorder have worse clinical and functional outcomes than women, and women are more likely to achieve symptom remission. Nevertheless, there is evidence that, despite women tending to exhibit good functioning after 1 year of treatment at early intervention services, this sex difference is not present after 2 years of treatment. It can be hypothesized that the sex differences seen in outcomes may be largely affected by other risk factors (premorbid functioning, age at onset of psychosis, attrition rate and so forth) [31]. On the other hand, there is evidence that women with schizophrenia spectrum disorders have more than double the odds of having physical comorbidities than men, particularly early in the course of psychiatric illness and at younger ages (<35 years) [32].

**Table 1 jpm-11-01279-t001:** Summary of included studies focusing on gender differences in schizophrenia and related psychotic disorders.

Study	Population	Study Type/Design	Results
González-Rodríguez et al., 2016 [15]	64 women with schizophrenia	12-week antipsychotic treatment	42 responders; time since menopause negatively associated with antipsychotic response
Spitz et al., 2017 [25]	126 at-risk and 94 FEP patients	Cross-sectional self-rated and clinician-rated assessment of positive and negative symptoms	Low agreement between self- and clinician-rated assessment; higher association with positive symptoms in FEP compared to at-risk and in women with respect to men, which became not significant after correction for multiple testing
Ittig et al., 2017 [22]	116 antipsychotic-naïve at-risk; 49 FEP patients	Blood sampling; prolactin assay	Hyperprolactinemia in 32% of at-risk and 35% of FEP patients; higher prolactin levels in women
Ferrari et al., 2018 [30]	25 FEP service users	Interview (in-depth) about help-seeking; cross-sectional	Women had more difficulties in accessing care; men had more difficulties in speaking about their symptoms
Egloff et al., 2018 [14]	65 at-risk (48 (73.8%) male; age = 25.1 ± 6.32) and 50 FEP (37 (74%) male; age = 27 ± 6.56) patients; 70 HCs (27 (38.6%) male; age = 26 ± 4.97)	3T MRI to test for reversal of sexual dimorphism of subcortical volumes in psychosis	Men had larger total brain (*p* < 0.001); smaller bilateral caudate (*p* = 0.008); smaller hippocampal (*p* < 0.001) volumes than women across all groups. Greater GM and WM volumes in older, compared to younger participants. No significant sex×group interaction
Menghini-Müller et al., 2019 [28]	336 at-risk participants (159 women) from 11 European centers	Cross-sectional assessment of clinical symptoms, drug use, comorbidity, and functioning	Men had more negative symptoms and current cannabis use than women; women scored higher on general psychopathology and more mood and anxiety comorbidity. No gender differences in global functioning. Correction for multiple testing abolished all significance
Dama et al., 2019 [31]	569 patients	Longitudinal study; demographics at baseline; positive, negative and total symptoms after 1 and 2 years	Men less educated and longer DUP, poorer premorbid functioning, more substance use, more negative symptoms, and poorer socio-occupational functioning than women. Women more remitting than men after 2 years of treatment and better functioning after 1 year. Results did not persist after adjustment for age at onset and confounders
Li, et al., 2019 [33]	360 FEP patients	Cross-sectional assessment of cognition through a comprehensive neurocognitive battery	Memory and working memory correlated with age at onset, negative symptoms, and adverse events in women. Processing speed correlated with antipsychotic dosage in men and adverse events in women. Selective attention correlated with reality distortion and negative symptoms in women, and age at onset and education in men. Executive function correlated with age at onset and reality distortion in women. All cognitive domains significantly correlated with educational level and functioning in both genders. Negative symptoms explained significant variability in cognition in both genders, while reality distortion, adverse events and mood symptoms differentiated between genders
Tong et al., 2019 [34]	60 female patients (mean age 24.57 ± 8.28) with recent-onset (5 years) psychotic spectrum disorder	Cross-sectional assessment of cognition through a cognitive battery, of perceived cognitive decline through the SCIS, and psychotic and depressive symptoms	SCIS not correlated with objective cognitive testing; SCIS positively correlated with depression and positive symptoms. Positive symptoms and depression together explained 31.9% of the total variance in SCIS; depression significantly predicted SCIS. Negative symptoms predicted memory test performance and correlated with it
Šimunović Filipčić et al., 2020 [32]	329 SSD patients; 837 HCs (general population)	Primary outcome: Prevalence of CPM	Women with SSD >3-fold odds for having CPM than men; in HCs, gender-related odds about the same. Prevalence of chronic disease in younger SSD women significantly higher than HCs, not in younger SSD men
Ayesa-Arriola et al., 2020 [35]	209 FEP patients (95 females and 114 males)	Clinical, cognitive, functioning, premorbid, and sociodemographic variables assessed at baseline (first contact), 1-, 3-, and 10-year follow-ups	At baseline, female age older than men, better premorbid adjustment, higher IQ, and better occupational status. Cannabis and alcohol consumption more frequent in men. During 1–3 years, women showed a better response to low antipsychotic doses and higher rates of recovery than men (50% vs. 30.8%). At 10-year follow-up, more females continued living independently and had partners, while schizophrenia diagnoses and cannabis consumption continued to be more frequent among men. Less negative symptoms in women
Rosen et al., 2020 [36]	152 male and 90 female referrals (88% at CHR; 35% converters)	Symptoms assessed cross-sectionally with the SIPS	All referrals: males, more negative and disorganization symptoms; among CHR converters: females, more perceptual abnormalities, bizarre thinking, and odd behavior; males, greater emotional blunting. Suspiciousness and disorganized communication predicted psychosis in males, trouble with focus and attention predicted psychosis in females
Studerus, et al., 2021 [23]	31 CHR (3 women), age 23.7 ± 4.82 years, 87 FEP patients (31 women), age 25.7 ± 9.15 years, and 45 HCs (21 women), age 28.0 ± 10.2 years from one Swiss (Basel) and one Spanish (Barcelona) center	Cross-sectional; blood withdrawal for prolactin assay and simultaneous completion of PSS	CHR and FEP patients scored higher PSS and had higher prolactin levels than HCs. Hyperprolactinemia in 26% of CHR and 45% of FEP patients. PSS scores correlated with mood symptoms. PSS scores did not correlate with prolactin levels. No gender-related differences in prolactin levels or perceived stress
Lähteenvuo et al., 2021 [37]	45,476 patients with schizophrenia aged <46 years at cohort entry, from one Swedish and one Finnish national cohort (more men in both cohorts, 58.2% and 52.5%, respectively; SUD, 30.90% Sweden [men, 70.4%], 26.28% Finland [men, 71.9%])	Longitudinal; SUD prevalence assessment (not smoking), Cox regression on risk of psychiatric hospitalization and all-cause and cause-specific mortality in SUD compared with no SUD during 22-year (1996–2017, Finland) and 11-year (2006–2016, Sweden) follow-ups	Similar results for the two Scandinavian cohorts. SUD prevalence 26% in Finland and 31% in Sweden. Multi-SUD, 4164 (48%) in Finland and 3268 (67%) in Sweden; AUD, 3846 (45%) in Finland and 1002 (21%) in Sweden; cannabis use disorder next greater frequency. SUD comorbidity associated with 50–100% increase in hospitalization and mortality with respect to no SUD. SUD more prevalent among men than among women
Irving et al., 2021 [29]	3350 FEP patients (62% males) at the South London and Maudsley NHS Trust with onset between 1 April 2007 and 31 March 2017	Cross-sectional; positive, negative, depressive, manic, and disorganization symptoms at initial clinical presentation; logistic regression	Poverty of thought, negative symptoms, social withdrawal, poverty of speech, aggression, grandiosity, paranoia and agitation more prevalent in men; tearfulness, low energy, reduced appetite, low mood, pressured speech, mood instability, flight of ideas, guilt, mutism, insomnia, poor concentration, tangentiality and elation more prevalent in women. Negative symptoms more common among men, depressive and manic symptoms more common among women even after adjusting for SUD
García-Rizo et al., 2021 [27]	491 drug-naïve FEP patients, aged 30.96 ± 9.96 years (213 women, aged 34.77 ± 10.80 years; 278 men, aged 28.03 ± 8.15)	Prolactin levels assessed through immunochemiluminescent automated assays to correlate with CRP, blood cell count, lipid and hepatic profile, and fasting glucose, stratified by sex	Prolactin was significantly correlated with CRP, LDL, AST in women and with HDL and eosinophil count in men. Women older than men. Prolactin levels higher in men than in women
Vázquez-Reyes et al., 2021 [26]	100 close relatives of patients with schizophrenia and related psychotic disorders; 64 men (64%), 36 women (36%); mean age during 2003–2007 38.26 ± 10.65; range = 18–65 years, during 2014-2017 51.42 ± 10.51; range = 30–77 years	Completion of SFS and the BPI during 2003–2007 and 2014–2017. Student’s *t*-test, ANOVA, and multivariate analysis of variance for comparisons of social functioning and behavior problems. Stepwise multiple linear regression analysis to predict the course of social functioning	No deterioration in social functioning or behavior problems. Women scored higher on withdrawal/social engagement, interpersonal behavior, independence-performance, independence-competence, and total social functioning, with no significant differences in behavior problems. Previous social functioning, underactivity/social withdrawal and education are predictive factors in the course of social functioning

Abbreviations: ANOVA, analysis of variance; AST, Aspartate Transaminase; AUD, alcohol use disorder; BPI, Behaviour Problems Inventory; CHR, clinical high risk for psychosis; CPM, chronic physical multi-morbidities; CRP, C-reactive protein; DUP, duration of untreated psychosis; FEP, first-episode psychosis; GM, gray matter; HCs, healthy controls; HDL, High-Density Lipoprotein; LDL, Low-Density Lipoprotein; MRI, magnetic resonance imaging; PSS, 10-item Perceived Stress Scale; SCIS, Subjective Cognitive Impairment Scale; SFS, Social Functioning Scale; SIPS, Structured Interview for Psychosis-Risk Syndromes; SSD, schizophrenia spectrum disorder; SUD, substance use disorder; T, Tesla; WM, white matter; ±, plus-minus standard deviation.

## 4. Discussion

We conducted a narrative review of sex-related differences in the psychoses, particularly schizophrenia, and found 17 articles to be eligible for further discussion, a summary of which may be found in Table 1. The fact that women often present with mood symptoms and tend to have better overall functioning at psychotic symptom onset compared with men may result in misdiagnosis and/or underestimation of their needs with a consequent delay in treatment and disadvantages in the subsequent course of illness [17].

Women are at a higher risk for metabolic and endocrine-induced side effects of antipsychotic drugs. This seems to be partly due to sex-specific pharmacokinetics (higher concentrations of free drug in target sites), enhancement of dopamine blockade by estrogen hormones, longer storage of antipsychotic drugs (due to the greater proportion of adipose tissue in women’s bodies), and higher risk of drug–drug interactions (because of women’s greater likelihood to being treated for comorbid illnesses) [18].

Besides the above, women are more likely to develop obesity, metabolic syndrome, cardiovascular diseases, and tardive dyskinesia after long-term treatment with antipsychotic drugs. This is probably due to the fact that women respond to acute stressors in a more pro-inflammatory manner compared to men, with increased immune response and decreased glucocorticoid sensitivity; in addition, gender difference of sex hormones profile and fluctuations, particularly during reproductive years, partly contribute to the increased burden of physical comorbidity observed in young female patients with a diagnosis of psychosis [38]. It has been noticed that chronic physical illnesses, although preventable and treatable, are the leading cause of premature mortality in patients with schizophrenia spectrum disorder [32].

Generally female patients tend to have better psychosis treatment outcomes and better control of physical comorbidities, with a consequent longer survival, because compared to men they tend to seek health care earlier and usually have more frequent clinical consultation rates. In such perspective, it has been also observed that young women with emerging psychosis have a higher correlation of self-rating with observer-rating regarding psychotic symptoms, and generally show better help-seeking behavior and are more partner-oriented when compared with men [25]. This may depend on greater insight of women compared to men and is confirmed by higher social functioning and less behavioral problems [26], but has the potential negative consequence of underestimating the real needs of female patients and jeopardizing engagement with psychiatric services. It has been noticed that, particularly for female patients with early psychosis, since subjective cognitive impairment seems to be significantly predicted by depression, treatments should not only focus on symptomatic remission and improved performance on cognitive tests, but also concentrate on improving mood and subjective cognitive function [34]. In a sample of adult-onset psychosis (defined as having an age of onset after 18 years of age, but in this particular sample, people whose psychosis emerged after the age of 25 years) the higher levels of correlation observed between female social functioning, negative symptoms and cognition, particularly executive function, suggest that these aspects are more interrelated that in men [33].

A very important issue is represented by differences in the long-term outcomes after a first episode of psychosis. As already reported, women show better outcomes, particularly during the first three years of treatment delivered at early intervention services, usually due to more favorable premorbid profiles and baseline characteristics. Moreover, the higher rates of mortality in patients with schizophrenia and comorbid substance use disorder (SUD) compared to those without SUD should be taken into account [37]. SUDs are dramatically oriented towards a male use, especially in adolescents and young adults presenting with psychotic symptoms [39]. This association between male gender and substance use can therefore represent the main reason explaining the worse outcome for males, at least at the beginning of the illness. However, not all adolescents who show some psychotic-like symptoms do convert into full-blown psychosis (for example, see Table 1, Ref. [36]). After an average period of ten years, outcomes for women tend to approximate those of men, while there is a general increase in dosage of antipsychotic medication once patients at first episode psychosis are transferred to community-based services. As a result, it can be argued that targeting sex differences, improving personalized approaches and prolonging the observation period at early intervention services should be considered a priority in treating psychosis [35]. A better collaboration between early intervention service and community-based service staffs could also help this goal.

Very often women spend more time than men in caregiving for infants and elderly relatives and consequently may not give priority to their own physical and mental health, so there is a need to program specialty team-based services for first-episode psychosis in close cooperation with primary care and pediatric services, in order to facilitate a timely and adequate access to care [17].

The present review contributes to confirm that both men and women inevitably have peculiar service needs for first episode and subsequent course of psychosis. Understanding gender-based specificity at first presentation, development and long-term progress of psychosis might facilitate better treatment orientation or identify patients especially likely to respond to a particular treatment [32]. The suggestion that women’s experiences in emotions are different from those of men, particularly but not only in cultures where gender roles and labels are still persistent, is basic for improving the study of gender-related issues and improving care programs that especially look at women’s mental health [34]. For example, it could be useful to take into account that women have a unique risk for developing psychosis in the peripartum period and traumatic events such as intimate partner violence, which affects more women than men, and is linked to an increased risk of psychotic experiences [17,40].

Furthermore, the possibility exists that men and women express differential relationships between psychopathology and cognition, and this could have remarkable implications for a better understanding of the neurobiological basis of psychosis and for future individualized interventions [33]. This would help not only to improve prediction of psychosis but also to deepen existing knowledge about possible sex-related differences in pathogenetic pathways and, consequently, to identify at-risk states [36]. Alongside further investigations of gender differences in illness behavior, coping, help seeking, compliance, psychopharmacology, hormone therapies, psychotherapy, and rehabilitation, there is a need for more methodologically sound, longitudinal, multidomain and interdisciplinary research focusing both on the sex (biological) and gender (psychosocial) perspective [41].

The transition to psychosis has been shown to occur according to a staging model much like the staging of the development of other medical disorders such as cancer and hypertension [42,43,44]. The staging has been proposed to be characterized by gender-specific factors [45,46]. Such factors have to be taken into account in implementing interventions aiming at improvement psychosis outcomes,

Another issue that should be taken into account is age. Age at onset of psychosis differs between males and females, with the latter presenting usually later, possibly also as a function of the protective role of the hormonal status of women [47]. Hence, those with earlier onset in females are likely to represent the most severe cases among them [48]. This further suggest that generalization is not always feasible, thus personalized care is the way forward.

## 5. Conclusions

Overall, from this review it is possible to conclude that gold standard care for psychosis should not only carefully consider different symptoms and specific characteristics of the two sexes but also addresses all the physiological, psychological, and social role needs of men and women suffering from this psychiatric disorder. Tailored treatment plans delivered by healthcare services should consider gender differences in prevalence, onset, clinical characteristics, treatment-response, outcome, comorbidities, aiming to deliver sex-specific prevention strategies, personalized care and effective outcomes in patients with a diagnosis of psychosis, with a particular attention to early phases of disease in the context of the staging model of psychosis onset.

## Figures and Tables

**Figure 1 jpm-11-01279-f001:**
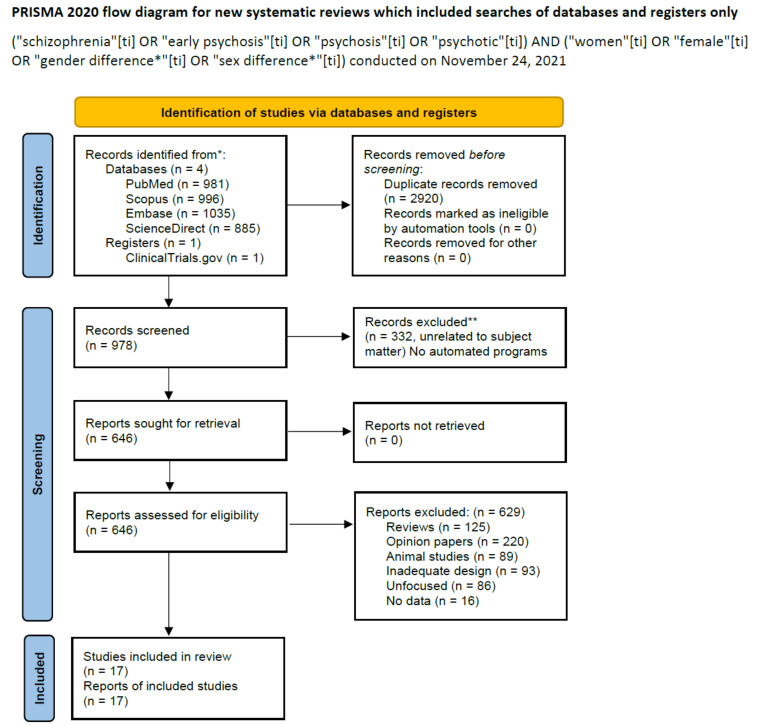
PRISMA Flow diagram of literature search, (http://www.prisma-statement.org/, accessed on 31st July 2021). * Consider, if feasible to do so, reporting the number of records identified from each database or register searched (rather than the total number across all databases/registers). ** If automation tools were used, indicate how many records were excluded by a human and how many were excluded by automation tools. From: Page MJ, McKenzie JE, Bossuyt PM, Boutron I, Hoffmann TC, Mulrow, CD, et al. The PRISMA 2020 statement: an updated guideline for reporting systematic reviews. BMJ 2021;372:Nn71. doi: 10.1136/bmj.n71.

## Data Availability

The data presented in this study are available on reasonable request from the corresponding author.

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
