# Peer review of "Psychosis in Women: Time for Personalized Treatment"

_jpm, 2021, doi:10.3390/jpm11121279_

Round 1

Reviewer 1 Report

These authors aim to show the need of a tailored treatment for psychosis regarding gender differences. This is a very important topic, which I totally support and agree with the authors that should be addressed. However, they do not address this concern in terms of medication types, quantitative and objective statements, nor measurable parameters in general; they just suggest gender differences obtained from the literature without supporting them with other than hypothesis and conclusions obtained from other groups, which in my opinion is quite scarce. If this is supposed to be a review of the latest data in the field, I strongly suggest the authors to include it and point it out from the beginning, in the abstract, as well as include a brief explanation regarding the methods they used to reach the conclusions they report and a significantly higher volume of references.

In my opinion, the material and methods used are described without detail, and this should be improved. Moreover, no statistical analyses were carried out at any point, and only a description of the databases search and the inclusion and exclusion criteria are pointed out. I would strongly recommend the authors to compile this information in a more appropriate way, such as adding tables, schemes, or any other visual tool to help the reader to see an overview of both the search parameters, results obtained and data treatment. From my point of view, the absence of comparative graphs and tables makes the paper weak. Furthermore, the authors explain that they found a very high volume of publications regarding this matter (4616 titles), but only 26 were chosen and added to the discussion. In my opinion, this number of papers described is definitely not enough, and they do not explain at any point why or how 4590 titles were discarded from the “analysis”. It is also noticeable the absence of references in several parts of the introduction, as well as lack of general description of objective numbers supporting their statements. Again, the aim of the work is not well defined, and I suggest the authors to highlight this point, as well as to add many more references, add at least some descriptive and comparative tables and graphs in the results section and discuss the data obtained in a proper scientific way.

The results section is not showing any result per se, any number, any analysis, and again, any graph or table. To me, it seems like a continuation of the introduction, not adding any new result to the scientific community. They also mention some hypothesis and discuss some of the parameters already discussed in other works, but they do not present any objective data. I strongly suggest to revisit this part of the paper, and transfer most of the information to either the introduction or the discussion, and add objective data analysis to this section.

The discussion is appropriate given the results they include but, in my opinion, they are not supporting the findings they report with enough objective data.

The authors have strong English skills, although there are some typos to take care of. A major weakness of this work is the lack of objective data, graphs and tables, as well as a little amount of papers included for the discussion.  

 To sum up, there is a need of general improvement, as I abovementioned. I would recommend the authors to 1. Add more references, 2. Analyze the data in an objective way and 3. Re-write the paper in a more logical and scientific way.

Author Response

Rome, November 24, 2021.

Dear Editor, we addressed the plagiarism issues, which were mostly trivial and ill-posed. We also addressed Reviewers’ issues and recommendations, using the Word tracking system. We also added text to the required 3,000 word count. We here respond point-to-point to Reviewers’ comments, keeping their original observations in this response, responding just underneath each point raised. We hope the new version will suit Reviewers’ requirements and be suitable for publication in the Journal of Personalized Medicine. We thank you and the reviewers for the effort put on the revision process.

REVIEWER 2

This manuscript focuses on the important issue of gender in psychosis.

The authors discuss the risk of women being diagnosed at the onset of psychotic symptoms.

We thank Reviewer for recognizing the importance of the issue we tackled. The risk was just one aspect of our aim.

"The prevalence of psychological distress among Japanese high school students ranges from approximately 10.7% (2007) to 7.6% (2013), and has been found to be related to household income, low parental education, and single parenting (Kachi et al.) These rates suggest that it is rather common for Japanese high school students to experience psychological problems. This suggests that it is rather common for Japanese high school students to experience psychological problems." (Okuyama et al. 2017) It should be added that a significant proportion of human life experiences psychosis during adolescence, and the majority of these do not transition into mental disorders.

We thank Reviewer for these considerations. However, we were unable to cite this precious literature, because it is not focused on psychosis. We added the concept that not all prodromal adolescents convert to psychosis.

Ref)Okuyama et al. School-based interventions aimed at the prevention and treatment of adolescents affected by the 2011 Great East Japan Earthquake: a three-year longitudinal study The Tohoku journal of experimental medicine, 242, 3, 203-213.

We thank Reviewer for letting us know. We could not add this important reference in our discussion, since it regarded PTSD and not psychosis.

I think this manuscript is an important review that examines gender differences in response to psychosis.

We thank Reviewer for this appreciation.

However, since age is important in psychosis, I would have liked to have added a discussion on age.

We thank Reviewer for the suggestion and added a discussion of age-related differences in psychosis as related to gender. We thank Reviewer for observations and suggestions that prompted us to improve our manuscript.

Reviewer 2 Report

This manuscript focuses on the important issue of gender in psychosis.

The authors discuss the risk of women being diagnosed at the onset of psychotic symptoms.

"The prevalence of psychological distress among Japanese high school students ranges from approximately 10.7% (2007) to 7.6% (2013), and has been found to be related to household income, low parental education, and single parenting (Kachi et al.) These rates suggest that it is rather common for Japanese high school students to experience psychological problems. This suggests that it is rather common for Japanese high school students to experience psychological problems." (Okuyama et al. 2017) It should be added that a significant proportion of human life experiences psychosis during adolescence, and the majority of these do not transition into mental disorders.

Ref)Okuyama et al. School-based interventions aimed at the prevention and treatment of adolescents affected by the 2011 Great East Japan Earthquake: a three-year longitudinal study The Tohoku journal of experimental medicine, 242, 3, 203-213.

I think this manuscript is an important review that examines gender differences in response to psychosis.

However, since age is important in psychosis, I would have liked to have added a discussion on age.

Author Response

Rome, November 24, 2021.

Dear Editor, we addressed the plagiarism issues, which were mostly trivial and ill-posed. We also addressed Reviewers’ issues and recommendations, using the Word tracking system. We also added text to the required 3,000 word count. We here respond point-to-point to Reviewers’ comments, keeping their original observations in this response, responding just underneath each point raised. We hope the new version will suit Reviewers’ requirements and be suitable for publication in the Journal of Personalized Medicine. We thank you and the reviewers for the effort put on the revision process.

REVIEWER 1

These authors aim to show the need of a tailored treatment for psychosis regarding gender differences. This is a very important topic, which I totally support and agree with the authors that should be addressed. However, they do not address this concern in terms of medication types, quantitative and objective statements, nor measurable parameters in general; they just suggest gender differences obtained from the literature without supporting them with other than hypothesis and conclusions obtained from other groups, which in my opinion is quite scarce. If this is supposed to be a review of the latest data in the field, I strongly suggest the authors to include it and point it out from the beginning, in the abstract, as well as include a brief explanation regarding the methods they used to reach the conclusions they report and a significantly higher volume of references.

We thank Reviewer for endorsing our viewpoint and for the consideration. We agree with the Reviewer that we did not address this concern in terms of medication types, quantitative and objective statements, nor measurable parameters in general; however, reviews are based on results obtained from the literature and hypotheses emerge as a result of the results and the conclusions of papers in literature which are based on work of other groups. We added text to respond to your concerns.

In my opinion, the material and methods used are described without detail, and this should be improved. Moreover, no statistical analyses were carried out at any point, and only a description of the databases search and the inclusion and exclusion criteria are pointed out. I would strongly recommend the authors to compile this information in a more appropriate way, such as adding tables, schemes, or any other visual tool to help the reader to see an overview of both the search parameters, results obtained and data treatment. From my point of view, the absence of comparative graphs and tables makes the paper weak. Furthermore, the authors explain that they found a very high volume of publications regarding this matter (4616 titles), but only 26 were chosen and added to the discussion. In my opinion, this number of papers described is definitely not enough, and they do not explain at any point why or how 4590 titles were discarded from the “analysis”. It is also noticeable the absence of references in several parts of the introduction, as well as lack of general description of objective numbers supporting their statements. Again, the aim of the work is not well defined, and I suggest the authors to highlight this point, as well as to add many more references, add at least some descriptive and comparative tables and graphs in the results section and discuss the data obtained in a proper scientific way.

We agree with the Reviewer that the Material and Methods section was quite lacking information. We added methods to describe what we did. We added Inclusion/Exclusion criteria, we tabulated some of our material to help readers to better understand what we did and found, and added details on exclusion. Please note that the 4616 titles obtained was wrong, as it regarded an unfocused search. We updated the search strategy and found only 978 articles to consider. We added reasons for exclusion and a PRISMA figure to meet the graphic/Table requirement. We added literature in Introduction to better support our theoretical background and better defined our aim. Statistical analyses were not possible due to the heterogeneity of included papers and were not the aim of this narrative review.

The results section is not showing any result per se, any number, any analysis, and again, any graph or table. To me, it seems like a continuation of the introduction, not adding any new result to the scientific community. They also mention some hypothesis and discuss some of the parameters already discussed in other works, but they do not present any objective data. I strongly suggest to revisit this part of the paper, and transfer most of the information to either the introduction or the discussion, and add objective data analysis to this section.

It is true, but this is a narrative review. However, we provided new text to address these issues. We followed your suggestions and hope to have met some of your requirements.

The discussion is appropriate given the results they include but, in my opinion, they are not supporting the findings they report with enough objective data.

We rewrote much of the Discussion section. We attempted at supporting findings with objective data and hope to have succeeded.

The authors have strong English skills, although there are some typos to take care of. A major weakness of this work is the lack of objective data, graphs and tables, as well as a little amount of papers included for the discussion. 

We thank reviewer for appreciation of our English skills and carefully controlled for typographic errors. The rest of the criticism in this point raised regards observations already made above and responded to. We repeat that we added text in Introduction, Methods, Results and Discussion and hope to have met your requirements.

To sum up, there is a need of general improvement, as I abovementioned. I would recommend the authors to 1. Add more references, 2. Analyze the data in an objective way and 3. Re-write the paper in a more logical and scientific way.

We thank Reviewer for the useful suggestions, that helped us to produce a better manuscript. We added references, looked at data and analysed them objectively, and rewrote much of the paper. We hope that the revised version will meet your taste.

Round 2

Reviewer 1 Report

Thank you for adding methods to describe the work carried out, especially the PRISMA figure, I think it is clear, concise and really useful. The literature added in the Introduction improves significantly the quality of the paper. I agree after reading the response that statistical analyses were beyond the scope of this narrative review. Thank you for your work.